# Wiz binds active promoters and CTCF-binding sites and is required for normal behaviour in the mouse

Luke Isbel[1], Lexie Prokopuk[1†], Haoyu Wu[2], Lucia Daxinger[1,2], Harald Oey[1‡], Alex Spurling[1], Adam J Lawther[3,4], Matthew W Hale[3,4], Emma Whitelaw[1*]

[1]Department of Biochemistry and Genetics, La Trobe Institute for Molecular Science, Melbourne, Australia; [2]Department of Human Genetics, Leiden University Medical Centre, Leiden, The Netherlands; [3]Department of Psychology and Counselling, La Trobe University, Melbourne, Australia; [4]School of Psychology and Public Health, La Trobe University, Melbourne, Australia

**Abstract** We previously identified *Wiz* in a mouse screen for epigenetic modifiers. Due to its known association with G9a/GLP, Wiz is generally considered a transcriptional repressor. Here, we provide evidence that it may also function as a transcriptional activator. Wiz levels are high in the brain, but its function and direct targets are unknown. ChIP-seq was performed in adult cerebellum and Wiz peaks were found at promoters and transcription factor CTCF binding sites. RNA-seq in *Wiz* mutant mice identified genes differentially regulated in adult cerebellum and embryonic brain. In embryonic brain most decreased in expression and included clustered protocadherin genes. These also decreased in adult cerebellum and showed strong Wiz ChIP-seq enrichment. Because a precise pattern of protocadherin gene expression is required for neuronal development, behavioural tests were carried out on mutant mice, revealing an anxiety-like phenotype. This is the first evidence of a role for Wiz in neural function.

**\*For correspondence:**
e.whitelaw@latrobe.edu.au

**Present address:** [†]Centre for Genetic Diseases, Hudson Institute of Medical Research, Melbourne, Australia; [‡]Translational Research Institute, University of Queensland Diamantina Institute, Brisbane, Australia

**Competing interests:** The authors declare that no competing interests exist.

## Introduction

An ENU mutagenesis screen for modifiers of epigenetic reprogramming was carried out in the mouse and the lines produced are termed *MommeD*s, *Modifiers of murine metastable epialleles, Dominant* (*Daxinger et al., 2013*). The screen used a multicopy GFP transgene, under the control of erythroid promoter/enhancer sequences, as a reporter. As this reporter undergoes stochastic silencing among otherwise identical erythroid cells, it is reminiscent of the position effect variegation (PEV) screens used to identify genes with roles in regulating gene expression (*Henikoff, 1990*; *Fodor et al., 2010*). Male mice with the reporter were treated with N-ethyl-N-nitrosourea (ENU) and offspring were screened for changes in the percentage of erythrocytes expressing GFP; this was a screen for dominant effects. The mutation underlying the *MommeD30* strain was found to be a single base pair deletion in the gene *Widely interspaced zinc finger motifs, Wiz*. The deletion, of an adenine, causes a frame-shift that introduces a premature stop codon. $Wiz^{MommeD30/MommeD30}$ embryos die around mid-gestation and heterozygosity for the mutant allele results in approximately half the normal levels of the protein, confirming that $Wiz^{MommeD30}$ is a null allele (*Daxinger et al., 2013*). This is the only mouse strain with a mutant form of this gene to be reported.

Little is known about the role of Wiz in any tissue. The protein is predicted to have 11 C2H2 type zinc finger domains that are unusually widely spaced. Some of them can bind DNA and some of them can mediate protein-protein interactions with the G9a/GLP (G9a-like) histone

methyltransferase complex (*Ueda et al., 2006*; *Bian et al., 2015*). The G9a/GLP heterodimer catalyses the methylation of H3K9me1 and H3K9me2 (*Jenuwein et al., 1998*; *Tachibana et al., 2001*, *2002*, *2005*), which is associated with transcriptional repression (*Barski et al., 2007*) and is located on chromatin in broad regions, termed LOCKs (large organized chromatin K9-modifications) (*Wen et al., 2009*). Work carried out in cell lines has shown that Wiz can stabilize the G9a/GLP complex, mediate its localisation to DNA and is required for maintaining normal global levels of H3K9me2, suggesting a role in transcriptional repression (*Bian et al., 2015*; *Simon et al., 2015*; *Ueda et al., 2006*).

It has also been shown that G9a can act as a transcriptional activator, independent of its histone methytransferase activity (*Chaturvedi et al., 2009*; *Lee et al., 2006*; *Oh et al., 2014*; *Purcell et al., 2011*; *Yuan et al., 2007*) and it has recently been suggested that WIZ has a role in the activation function of G9a (*Simon et al., 2015*). $Wiz^{MommeD30}$ was identified in the screen because mice heterozygous for the mutant allele showed decreased expression of the GFP transgene. Here, we show at an independent locus known to be sensitive to the dosage of epigenetic modifiers, the A*gouti viable yellow (A$^{vy}$)* allele, that haploinsufficiency for Wiz also results in decreased expression.

Wiz is expressed highly in the brain, both in the embryo and in the adult (*Matsumoto et al., 1998*). As reduced levels of G9a/GLP are known to result in the neurological disease Kleefstra syndrome (*Kleefstra et al., 2006*; *Willemsen et al., 2011*), it is possible that reduced levels of Wiz might have neurological consequences. Kleefstra syndrome individuals display a complex range of neurological symptoms, including anxiety (*Willemsen et al., 2011*). We decided to study the role of Wiz in the brains of $Wiz^{MommeD30}$ mice. We carried out ChIP-seq and RNA-seq in adult cerebellum, a tissue in which Wiz is expressed at high levels (*Matsumoto et al., 1998*). Our results show that Wiz binds tens of thousands of transcriptional regulatory regions across the genome, some of which decrease in expression in $Wiz^{MommeD30/+}$ tissue. ChIP-seq, with an anti-Wiz antibody, has been reported in a human kidney cell line (*Bian et al., 2015*) and there is little overlap with our findings. Our molecular studies prompted us to assess behaviour in heterozygous mutant mice and we found that $Wiz^{MommeD30/+}$ mice display an anxiety-like phenotype.

## Results

### Wiz haploinsufficiency increases the probability of silencing the metastable epiallele, *Agouti viable yellow*

Haploinsufficiency for Wiz resulted in an increase in the probability of silencing at the GFP transgene reporter, suggesting a role in transcriptional activation at this locus (*Daxinger et al., 2013*). To test the effect of haploinsufficiency for Wiz at an independent reporter known to be sensitive to the dosage of epigenetic modifiers, we used mice carrying the A$^{vy}$ allele. This strain has been maintained on the C57BL/6J background in a heterozygous state (A$^{vy}$/a). The coat colour of the mouse can be used as a reporter of transcriptional activity at the A$^{vy}$ locus (*Duhl et al., 1994*). An active locus results in a yellow coat, a silent locus results in a dark brown coat (called pseudoagouti) and a mottled coat arises when some clusters of cells express the locus and some clusters do not. Male $Wiz^{MommeD30/+}$ mice (maintained on an FVB/NJ background) were crossed to female A$^{vy}$/a (pseudoagouti) mice and the coat colours of the offspring were scored at weaning (three weeks). FVB/NJ mice carry a wildtype *Agouti* locus, A/A, so ~50% of offspring from these crosses will be A$^{vy}$/A and differ only at the $Wiz^{MommeD30}$ allele, i.e. $Wiz^{MommeD30/+}$; Avy/A or $Wiz^{+/+}$; Avy/A. Offspring were genotyped for the A$^{vy}$ and $Wiz^{MommeD30}$ alleles. Mice that did not inherit the A$^{vy}$ allele we excluded from the analysis. The reciprocal cross was also carried out, i.e. female $Wiz^{MommeD30/+}$ mice were crossed to male A$^{vy}$/a (pseudoagouti) mice. The pedigrees and outcomes for both crosses are shown in *Figure 1A and B*.

$Wiz^{MommeD30/+}$ offspring from both crosses displayed a shift towards silencing (i.e. more pseudoagouti mice), compared to $Wiz^{+/+}$ littermates. The shift was statistically significant in both cases; the shift from $Wiz^{MommeD30/+}$ sire cross had a p-value of 0.0002, and from the $Wiz^{MommeD30/+}$ dam cross had a p-value of 0.0458 (*Figure 1A and B*). The effect of haploinsufficiency at the A$^{vy}$ locus is similar to the effect at the GFP transgene reporter, i.e. reduced levels of Wiz were associated with increased silencing.

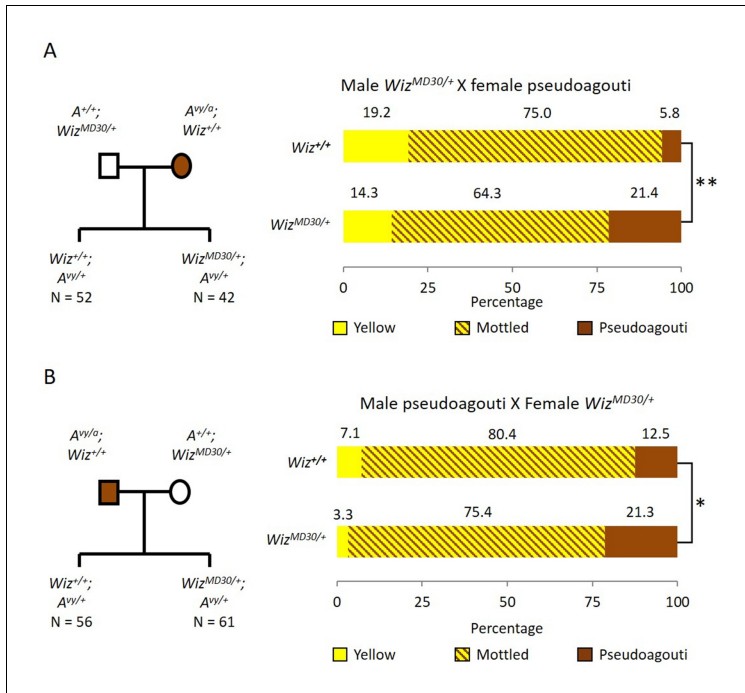

**Figure 1.** Wiz haploinsufficiency increases silencing at the $A^{vy}$ locus. Pedigree charts (left) are shown for generating F1 mice heterozygous for the $A^{vy}$ allele and either wildtype or heterozygous for $Wiz^{MommeD30}$. $Wiz^{MommeD30/+}$ mice were crossed to pseudoagouti $A^{vy}/a$ mice, numbers of F1 mice in each cohort are indicated. The proportions of coat colours for F1 mice are shown (right) from a cross with either a $Wiz^{MommeD30/+}$ sire (**A**) or dam (**B**). Chi-squared tests were carried out to determine significance, *p-value<0.05, **p-value<0.005.

## Wiz peaks occur at regulatory regions, many of which bind CTCF

To test the specificity of an anti-Wiz antibody (NBP180586, Novus Biologicals, Littleton, USA) in mouse tissue, Western blotting was carried out using total protein lysates from $Wiz^{+/+}$ and $Wiz^{MommeD30/MommeD30}$ E12.5 embryonic heads, prior to the death of the $Wiz^{MommeD30/MommeD30}$ fetuses (*Figure 2—figure supplement 1*). The bands detected in the wildtype sample (at around 100 kDa) were absent in the sample from the homozygous mutant, consistent with an antibody that is specific for Wiz. A larger ~160 KDa isoform has previously been reported in adult cerebellum (*Matsumoto et al., 1998*), and consistent with that report, a band of this size is detected with our antibody in protein extracted from this tissue (*Figure 2—figure supplement 1*). In addition, we used an unbiased tandem affinity purification approach and compared mass spectroscopy data with previously published datasets analysing Wiz binding partners. Co-immunoprecipitation (Co-IP) with the anti-Wiz antibody and an anti-IgG control antibody was carried out on pooled E13.5 brain tissue (n = 1 pooled sample from 5 individuals) and adult cerebellum (n = 2 biological replicates). Four members of the Wiz-Zfp644-EHMT1-EHMT2 complex (*Ueda et al., 2006*; *Bian et al., 2015*) were in the top five proteins in the dataset, ranked by peptide count in embryonic brain (*Figure 2—source data 1*).

The cerebellums of four adult male $Wiz^{+/+}$ mice were pooled for ChIP-seq; two cerebellums per sample and two biological replicates. The cerebellum was chosen for two reasons; it is a relatively simple tissue in the brain with respect to cell types and it is an ENCODE tissue, enabling comparisons to be made with ChIP-seq data obtained for other chromatin proteins. Library preparation and sequencing were carried out by Active Motif (Carlsbad, USA). Over 30 million reads were generated for each of the two Wiz ChIP-seq datasets and the chromatin Input sample dataset. At least 24 million reads could be aligned to the mouse mm10 genome in all cases, after filtering out PCR duplicates.

Approximately 40,000 peaks were identified at which the two Wiz-ChIP-seq datasets were significantly enriched compared to the Input control with a p-value of $\leq 1 \times 10^{-20}$ (*Supplementary file 1*).

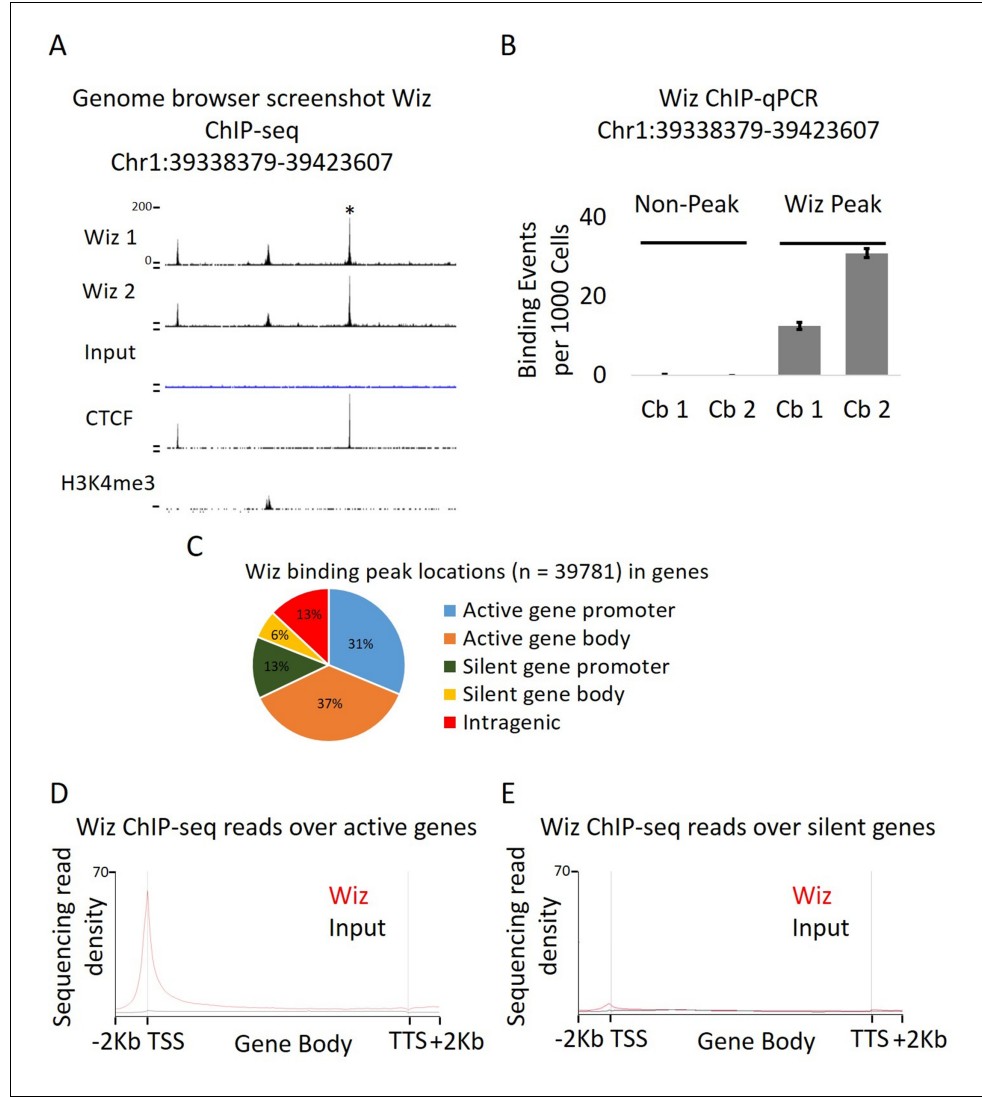

**Figure 2.** Wiz binds across the genome and at promoter elements. (**A**) ChIP-seq was performed for Wiz in the adult male cerebellum, a screenshot is shown of a random Wiz-enriched site (* indicated) with the two ChIP-seq replicates and input, from the ~40,000 significantly enriched peaks. Encode data for CTCF and H3K4me3 is also included. (**B**) Enrichment for Wiz in two cerebellum samples (Cb1 and Cb2) is shown by ChIP-qPCR, with primers located in regions not enriched for Wiz in ChIP-seq data and primers flanking the * indicated ChIP-seq peak from (**A**). Enrichment is represented as binding events detected per 1000 cells and are generated by running samples in parallel with known amounts of genomic DNA. Error bars indicate S.D. from 3 technical replicates. (**C**) The percentage is shown of Wiz peaks that overlap with the promoter (up to 2 kb from a TSS) and genic sequence of genes that were classified as active or silenced, according to RNA-seq data mapped to Ensembl genes annotations. (**D**) Wiz ChIP-seq (and Input) occupancy over all Ensembl gene bodies is shown as deep sequencing read density along the transcription unit, including 2 kb up and downstream of the transcriptional start and stop site. Genes were separated into either active or silenced transcriptional states, as in **C**.

The following source data and figure supplement are available for figure 2:

**Source data 1.** Anti-Wiz antibody co-immunoprecipitation.

**Figure supplement 1.** Anti-Wiz antibody western blotting.

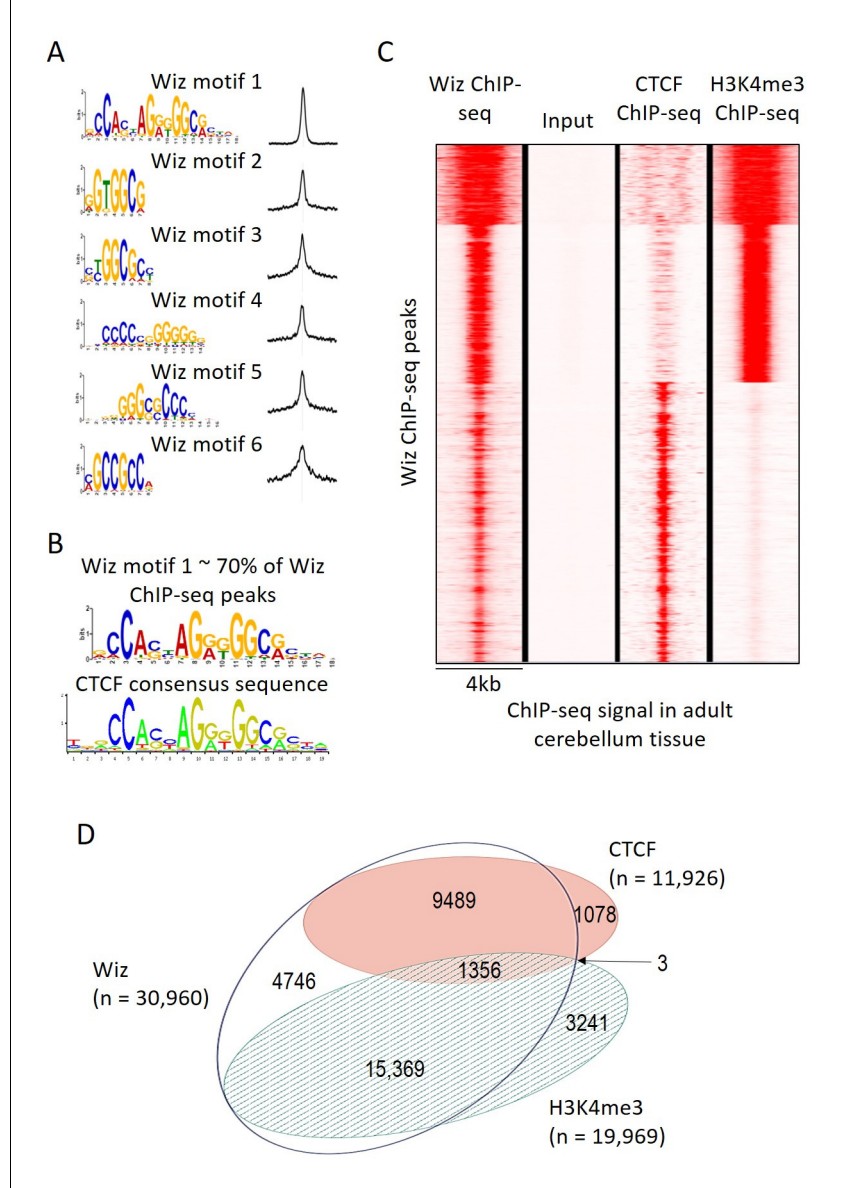

**Figure 3.** Wiz binding consensus shows a high degree of overlap with that for CTCF. (**A**) The six most enriched motifs in Wiz ChIP-seq peaks (n = 39,781) and their distribution inside peaks are shown. (**B**) The most significant of these matches the CTCF-binding site consensus sequence (JASPAR CORE database - MA0139.1). (**C**) Read density for Wiz and the Encode CTCF and H3K4me3 ChIP-seq datasets were calculated across a 4 kb region centred on the ~40,000 Wiz ChIP-seq peaks. Input sequencing from the Wiz ChIP-seq experiment is also shown. Loci are clustered by the similarity of read density and datasets were normalized to the smallest sized library. (**D**) Venn diagram showing the overlap for Wiz, CTCF and H3K4me3 ChIP-seq peaks. An overlap was defined as at least one base of sequence occupied by a significantly enriched peak (p≤1 × $10^{-20}$) from two datasets.

The following figure supplement is available for figure 3:

**Figure supplement 1.** Remapping publically available Wiz ChIP-seq data.

A representative Wiz peak and ChIP-qPCR validation at the same locus is shown in *Figure 2A and B*, respectively. To determine where Wiz was bound with respect to genic regions of transcriptionally active and inactive genes, an analysis of overlap with gene promoters and gene bodies was carried out. The peaks were roughly equally distributed across promoters (defined as 2 kb up and

downstream of the TSS), intragenic and intergenic regions, despite the fact that promoters comprise only a small fraction of the total mouse genome (*Figure 2C*). Consistent with this, the deep sequencing read density across the body of all Ensembl genes (normalized for gene length; see Materials and methods), as well as 2 kb upstream of the TSS (transcription start site) and downstream of the TTS (transcriptional termination site), showed strong enrichment at the TSS (*Figure 2D and E*).

The MEME-ChIP program was used to identify motifs that are common at Wiz-binding locations and several were identified with high statistical significance, i.e. p-values from $1.1 \times 10^{-6902}$ to $2.9 \times 10^{-439}$ (*Figure 3A*). One of these, Wiz motif 1, had a significance (MEME-ChIP E-value) E-Value of $2.3 \times 10^{-6903}$, an order of magnitude higher than other motifs and aligned almost perfectly with the CTCF-DNA binding motif (*Figure 3B*). This motif was present in approximately 70% of all Wiz binding peaks across the genome.

An ENCODE ChIP-seq dataset for CTCF binding in mouse adult cerebellum is publically available. To determine if CTCF peaks overlapped with Wiz peaks, Wiz ChIP-seq read density was compared to ENCODE ChIP-seq data for CTCF in the adult cerebellum at Wiz peaks (*Figure 3C*). Strong overlap was seen. The ENCODE dataset for H3K4me3, a histone modification associated with active transcription, was also mapped to the genome (mm10 build) and ChIP-seq peaks for CTCF, H3K4me3 and Wiz were compared (*Figure 3C*), as described in Materials and methods. Over half of Wiz ChIP-seq peaks overlapped the promoter mark H3K4me3 and approximately a third of all Wiz ChIP-seq peaks overlapped CTCF ChIP-seq peaks (*Figure 3D*). Virtually all locations with both CTCF and H3K4me3 peaks also showed a Wiz peak (1356/1359). Interestingly, almost all CTCF peaks overlapped with Wiz peaks.

A ChIP-seq dataset was recently published using an anti-Wiz antibody and HEK293T cells (*Bian et al., 2015*); the authors conclude that Wiz localizes to gene promoters and (using EMSA analysis) can bind to the DNA sequence CATTCCATTCCATT. Shown in their report are representative screen shots of Wiz-ChIP-seq enrichment at the promoter region of seven genes (defined in their study as 5 kb up or downstream of the TSS) where there was a "consensus DNA-binding motif confirmed at the locus". We examined the DNA sequence surrounding each of these seven genes, in all cases the location of the consensus sequence was at least 5 Kb from either the promoter where the binding was demonstrated or the coordinates of the nearest reported ChIP-seq peak (n = 11,853). Instead, the sequence was located downstream in the body of each gene. After remapping this publically available Wiz ChIP-seq dataset (GSE62616), only one of the seven genes was found to have enrichment at the consensus sequence (i.e. reads mapping directly at the CATTCCATTCCATT motif), in the *SENP5* gene. In this case, reads mapped to a sequence located in a repetitive HSATII satellite element and is likely to be a mapping artefact (*Figure 3—figure supplement 1*). Enrichment was also seen at other copies of the HSATII element across the genome that had the CATTCCAT-TCCATT sequence (data not shown). We were unable to reproduce the enrichment shown in the report at any of the seven gene promoters (data not shown).

## Genome-wide expression changes in embryonic brains and adult cerebellum of *Wiz^MommeD30/+* mice

Our results have shown that haploinsufficiency for Wiz alters expression at the GFP transgene reporter and the $A^{vy}$ locus. We were keen to see if haploinsufficiency for Wiz resulted in changes in the expression of other genes and chose to investigate this in the brain. Firstly, we used the whole brains of E13.5 embryos in an attempt to identify early events and secondly, we used adult cerebellum, to compare expression changes to Wiz ChIP-seq data.

Initially, RNA-seq was performed using E13.5 brains from male *Wiz^MommeD30/+* and *Wiz^+/+* embryos (n = 3 biological replicates for each genotype). Reads were aligned to the mouse genome (mm10 build) and differential expression of Ensembl genes between genotypes was calculated using the R package DESeq (*Anders and Huber, 2010*) (*Figure 4A*; *Supplementary file 2*).

RNA-seq was also carried out using brains from female E13.5 *Wiz^MommeD30/+* and *Wiz^+/+* embryos, and differential expression of genes was calculated as described above (n = 2 biological replicates for each genotype, *Figure 4A*; *Supplementary file 3*).

Genes with a fold-change in average expression of less than 0.7 or greater than 1.3 (shown in figures on log2 scale) and an adjusted p-value of ≤0.05, were considered significantly differentially expressed. A total of 36 were found to change in males and 44 changed in females. The majority of

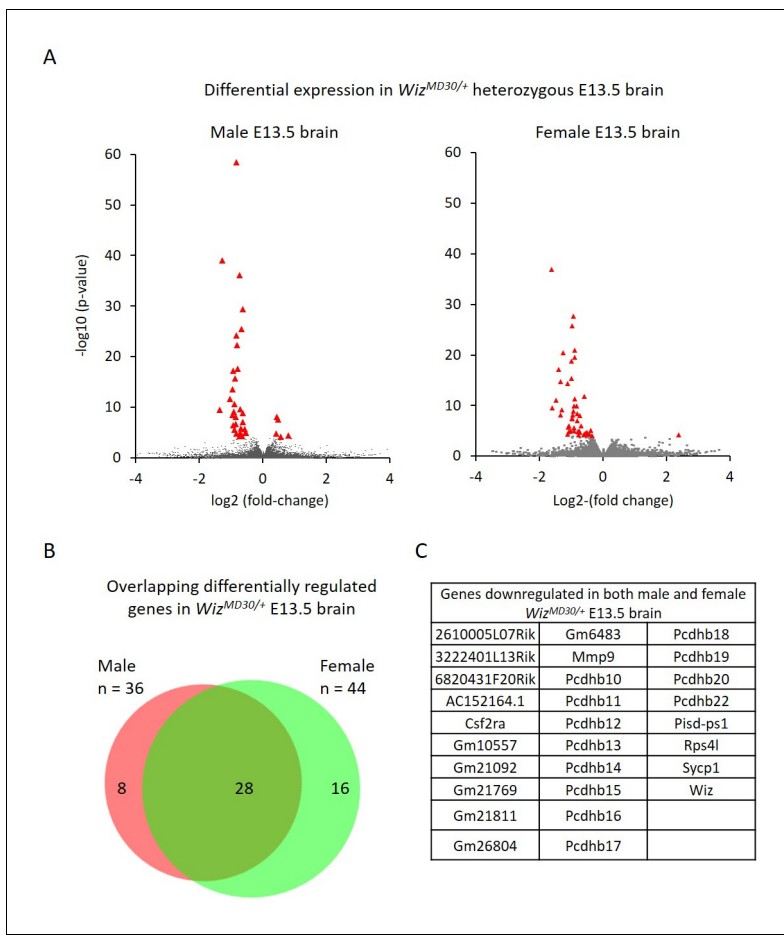

**Figure 4.** Differential gene expression in male and female *Wiz*$^{MommeD30/+}$ E13.5 brains. (**A**) Volcano plots show the log2 fold-change (x-axis) in the average expression of genes in *Wiz*$^{MommeD30/+}$ E13.5 brains compared to *Wiz*$^{+/+}$ E13.5 brains. Shown is data for males (left, n = 3 per genotype) and females (right, n = 2 per genotype). Significance is shown on a –log10 scale, those genes indicated in red are significantly differentially regulated with an adjusted p-value<0.05. (**B**) The overlap in differentially expressed genes is shown for the male and female embryonic brains, the majority of transcripts differentially expressed represent a common set between the sexes, listed in alphabetical order in (**C**), and all of these decrease in expression.
The following figure supplement is available for figure 4:

**Figure supplement 1.** Location of differentially expressed genes in E13.5 *Wiz*$^{MommeD30/+}$ brains.

significantly differentially expressed genes, in both cases, decreased in expression in the mutants (31/36 in males and 43/44 in females). As expected, *Wiz* itself was in this group.

Of the combined set of 80 gene expression changes in males and females, 28 genes were common between the sexes and all of these decreased in *Wiz*$^{MommeD30/+}$ samples compared to wildtype samples (*Figure 4B and C*). Genes that were significantly differentially expressed in one sex but not in the other generally trended towards statistical significance in the latter and none of these had a fold-change direction that was different between sexes, i.e. genes statistically down regulated in males trended down in females and vice versa.

Among the significantly differentially expressed genes, a relatively high proportion were from two gene clusters, the protocadherin β (Pcdhb) gene cluster on Chromosome 18 and a second cluster consisting of *2610005L07Rik, 6820431F20Rik, AC152164.1, Gm10557, Gm21092, Gm21769, Gm21811, Gm26804* and *Gm6483* on Chromosome 8 (*Figure 4—figure supplement 1*). The latter cluster is poorly annotated due to a nearby repeated region that has made mapping difficult

(**Boyle and Ward, 1992**). Under the annotations listed in the UCSC genome browser (**Karolchik et al., 2003**), most are annotated as cadherin11 pseudogenes; all have introns, and some are predicted to be coding. While it is very likely that Wiz haploinsufficiency affects multiple members of the cadherin-11 pseudogene cluster, it is possible that some expression changes are mapping artefacts. A more complete genome assembly is needed before this can be resolved by RNA-seq or single locus testing (i.e. real time quantitative PCR).

To see if the other significantly differentially expressed genes in E13.5 brain were related to location, we examined their locations on each chromosome (**Figure 4—figure supplement 1**). No other gene clusters were apparent, though it was noticed that several genes were located near telomeres and four of the significantly differentially expressed genes, *Samd11, psid-ps1, psid-ps2* and *Csf2ra*, are the last gene on the chromosome. While it is possible that the locations of these genes and the cadherin-11 pseudogene cluster are incorrect in the current genome build (mm10 build), they are likely to remain adjacent to heterochromatic repeats.

Cerebellums were also dissected from adult males (n = 3 per genotype) and RNA-seq was carried out on each. A total of 82 genes were significantly differentially expressed; 29 genes decreased and 53 increased in expression in $Wiz^{MommeD30/+}$ samples compared to $Wiz^{+/+}$ samples (**Figure 5A**; **Supplementary file 4**). Of these, some genes (n = 13) overlapped with those detected in the E13.5 brain data, including some from the protocadherin β cluster, some from the cadherin 11-like cluster, *Wiz, Csf2ra* and *Rps4l* (**Figure 5B**). These all decreased in expression in $Wiz^{MommeD30/+}$ cerebellum compared to the $Wiz^{+/+}$ cerebellum, as they had done in the embryonic brain.

In general, the down regulation of protocadherin β genes was more pronounced in the embryonic brain than in the adult cerebellum, where significance was seen for only a minority of the genes in the clusters (**Supplementary file 4**).

ChIP-seq data were investigated at the protocadherin β cluster (**Figure 6A**). A genome browser screenshot of the Wiz binding is shown for the protocadherin β cluster and its 5' enhancer (HS5-1bL). Wiz enrichment was detected at all protocadherin β promoters except *Pcdhb1*, which is the only protocadherin β gene that lacks a canonical protocadherin promoter sequence (**Guo et al., 2015**; **Wu et al., 2001**). The greatest enrichment was detected at the HS5-1bL enhancer. This enrichment at the enhancer is an order of magnitude higher than that seen at the individual promoters, and this might reflect the fact that while single members of the cluster are expressed in individual neurons, the HS5-1bL enhancer is likely to be active in all cell types (**Yokota et al., 2011**). In

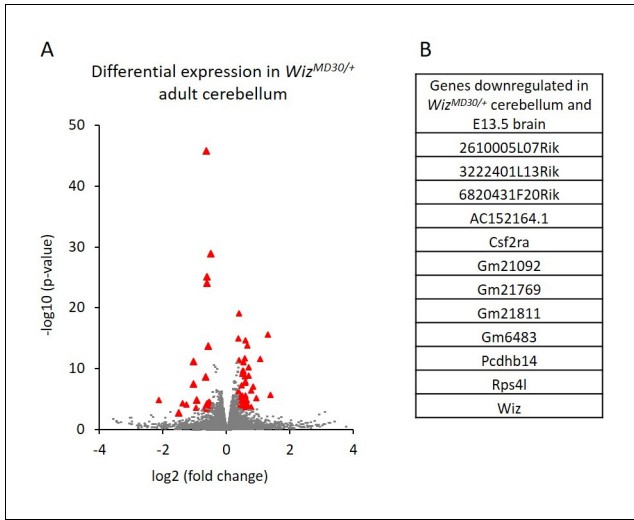

**Figure 5.** Differential gene expression in $Wiz^{MommeD30/+}$ adult cerebellum. (**A**) The log2 fold-change (x-axis) in the average expression of genes is shown for $Wiz^{MommeD30/+}$ and $Wiz^{+/+}$ cerebellum. Three male biological replicates were used per genotype. Significance is shown on a –log10 scale, those genes indicated in red are significantly differentially regulated. (**B**) A list of significantly regulated genes in $Wiz^{MommeD30/+}$ cerebellum that were also deregulated in E13.5 $Wiz^{MommeD30/+}$ brains.

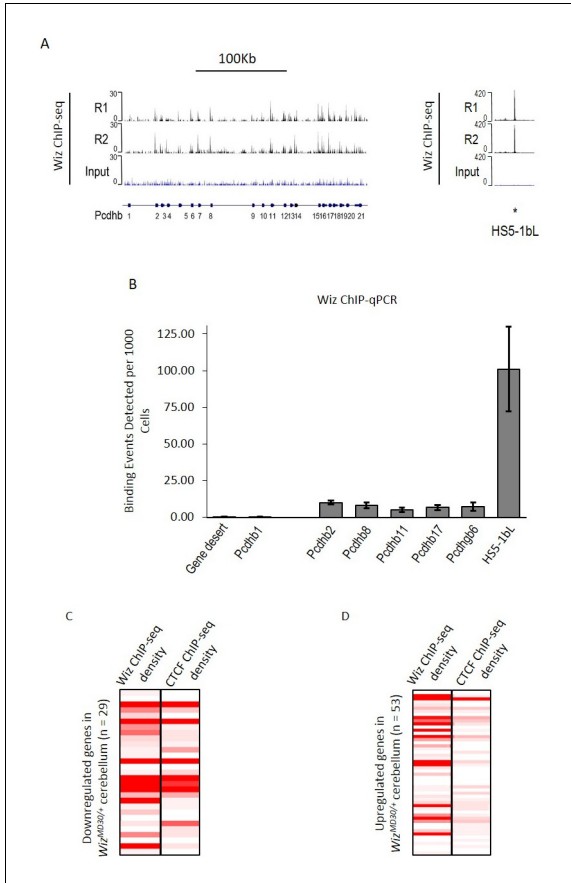

**Figure 6.** Wiz binding at the protocadherin β locus in the adult cerebellum. (**A**) Genome browser screenshots of the protocadherin β locus and the HS5-1bL enhancer (*indicated) located ~400 Kb downstream. Shown are Wiz ChIP-seq data generated in the adult cerebellum as the read depth along the locus. Wiz binds promoters of the Pcdhb locus (left) and is enriched more at the downstream HS5-1bL enhancer (right), the relative binding of Wiz is indicated by the altered scale bar. The location of the Pcdhb genes, are indicated. (**B**) ChIP-qPCR was performed using the anti-Wiz antibody for two negative Wiz-enrichment sites, with primers located in a gene desert and at *Pcdhb1*, which was not enriched in ChIP-seq, and six positive Wiz-enrichment sites, including the promoters of four Pcdhb genes, the promoter of a Pcdhg gene and the downstream HS5-1bL enhancer. Enrichment is represented as binding events detected per 1000 cells and are generated by running samples in parallel with known amounts of genomic DNA. Error bars are the SEM from 3 biological replicates. (**C**) Shown is Wiz ChIP-seq read density in adult cerebellum as a heatplot for the 29 genes that decreased and (**D**) the 53 genes that increased, in expression in the cerebellum of $Wiz^{MommeD30/+}$ mice. Also shown is CTCF ChIP-seq read density. ChIP-seq density was calculated for the 1 Kb surrounding the TSS of each gene and genes are ranked from the lowest to the highest fold change in each case. Datasets were normalized to the size of smallest library.

all cases tested, enrichment of Wiz was validated using ChIP-qPCR at protocadherin β promoters and the HS5-1bL element (*Figure 6B*). Conversely, primers located in a gene desert and the *Pcdhb1* promoter, which did not show an expression change in any of the tissues tested, were not enriched for Wiz.

Wiz and CTCF ChIP-seq read density were calculated at the transcriptional start sites of the ~82 significantly differentially regulated genes in the cerebellum. The genes were first separated by whether they decreased or increased in expression in $Wiz^{MommeD30/+}$ cerebellum compared to $Wiz^{+/+}$ cerebellum. The genes that decreased in expression showed high enrichment for Wiz and some enrichment for CTCF at the TSS (*Figure 6C*), while genes that increased in expression showed little enrichment for Wiz and no enrichment for CTCF (*Figure 6D*), suggesting that those showing increased expression were secondary effects (see Discussion).

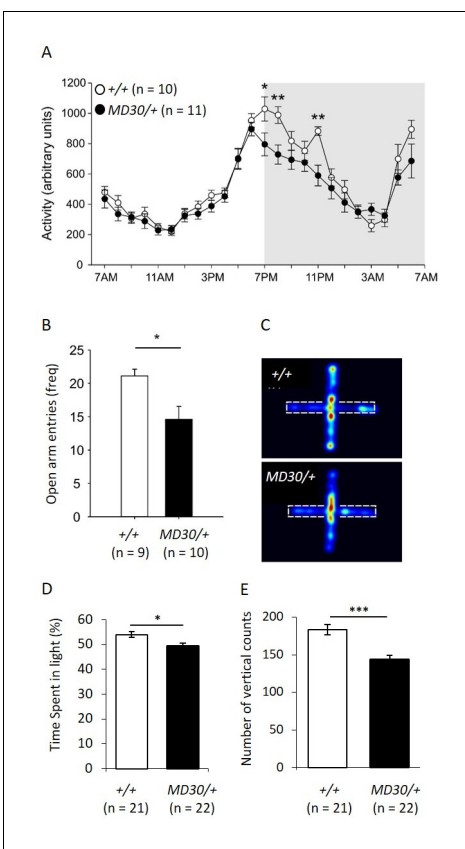

**Figure 7.** $Wiz^{MommeD30/+}$ mice show decreased activity and increased anxiety-like behaviour. (**A**) Line graph illustrating the average 24 hr activity profile from $Wiz^{+/+}$ and $Wiz^{MommeD30/+}$ mice, collapsed from 12 days of testing. The pattern of activity is broadly similar between the genotypes, though $Wiz^{MommeD30/+}$ mice show decreased activity in the first two hours and the fifth hour of the active phase of the light/dark cycle. Grey box indicates the dark phase of the light/dark cycle. (**B**) Bar graph showing decreased open arm entries among $Wiz^{MommeD30/+}$ mice compared with $Wiz^{+/+}$ mice, after 5 min intervals on the EPM test. (**C**) Heat map of locomotor activity on the EPM from a representative $Wiz^{+/+}$ mouse (top) and a representative $Wiz^{MommeD30/+}$ mouse (bottom). Broken lines indicate the location of the closed arms of the EPM. (**D**) Percentage of time $Wiz^{+/+}$ (n = 21) and $Wiz^{MommeD30/+}$ (n = 22) mice spent in the light portion of a light-dark test. (**E**) The number of vertical counts (rearing) $Wiz^{+/+}$ and $Wiz^{MommeD30/+}$ mice showed in the light-dark test. Error bars represent ± SEM, *p-value <0.05, **p-value 0.005, ***p-value <0.0005.

The following figure supplement is available for figure 7:

**Figure supplement 1.** No difference in temperature in $Wiz^{MommeD30/+}$ mice.

## $Wiz^{MommeD30/+}$ mice have altered behaviour

To characterize the behavioural phenotype of $Wiz^{MommeD30/+}$ mice, singly housed males were monitored over a period of 12 days using telemetry devices that record body temperature and locomotor activity. An initial cohort of $Wiz^{+/+}$ and $Wiz^{MommeD30/+}$ age-matched mice (n = 5 per genotype) was tested and a second cohort (n = 5 $Wiz^{+/+}$, 6 $Wiz^{MommeD30/+}$) was tested two months later to validate the results from the first. The data from both cohorts showed similar effects and were combined for presentation. Mice normally display a circadian pattern of activity dependent on the light/dark cycle and so data is shown across the 24-hr period after averaging results at each time point across the 12 days. No shift was seen in the circadian pattern, but decreased activity was seen in the first two hours and the fifth hour after lights off in $Wiz^{MommeD30/+}$ mice compared with the $Wiz^{+/+}$ mice (*Figure 7A*; time x genotype interaction, $F_{(23, 391)} = 3.923$, p<0.001, $\varepsilon = 0.350$).

Temperature profiles were also collected for each of the mice by collapsing three days of monitoring into hourly intervals and taking the average from each 24-hr time point. No difference was seen in $Wiz^{MommeD30/+}$ mice compared to $Wiz^{+/+}$ mice at any of the time points (*Figure 7—figure supplement 1*).

The elevated plus maze (EPM) is a test of anxiety-like behaviour in mice (*Walf and Frye, 2007*). The same mice used for telemetry monitoring were subjected to an EPM test following the 12-day monitoring period. $Wiz^{MommeD30/+}$ showed an approximately 25% reduction in the frequency of entry into the open arms of the EPM apparatus compared with the $Wiz^{+/+}$ mice (*Figure 7B and C*; t(17) = 2.90, p=0.010). Importantly, mice were assayed during 'lights on' periods (around 2pm - 4pm) when activity was not different between the strains (*Figure 7A*).

To validate the EPM data, two independent cohorts of mice ($Wiz^{MommeD30/+}$, n = 22; $Wiz^{+/+}$, n = 21) were subjected to a light-dark box test of anxiety-like behaviour. $Wiz^{MommeD30/+}$ mice showed reduced time spent in the light and reduced rearing activity (*Figure 7D and E*). These findings are consistent with an anxiety-like phenotype in $Wiz^{MommeD30/+}$ mice.

## Discussion

### *Wiz* is involved in transcriptional activation and binds many promoters in neural tissue

We previously described a *Wiz* mutation in the *MommeD30* strain (*Daxinger et al., 2013*). This was the only exonic mutation within the linked interval and in agreement with this being the causative *MommeD30* mutation, RNA-seq data presented here show no other genes within the linked interval were miss-expressed in mutants.

While a role for Wiz in transcriptional repression is well recognized (*Mulligan et al., 2008*), the notion that it may be involved in transcriptional activation is less well documented. This study supports the latter idea in a number of ways. Firstly, we show that haploinsufficiency for Wiz results in reduced expression at the $A^{vy}$ allele and previous work from our laboratory has shown that half the normal levels of Wiz is associated with reduced expression of a multicopy GFP transgene (*Daxinger et al., 2013*). By focussing on neural tissue, we have gone on to show that haploinsufficiency for Wiz results in decreased expression of a number of genes, mainly within gene clusters or adjacent to telomeric repeats in both embryonic brain and adult cerebellum. While any instance of decreased expression could, theoretically, be an indirect effect, our ChIP-seq data suggests otherwise. At most of the sites at which decreased expression is observed, Wiz peaks were detected. Furthermore, analysis of the set of genes with altered expression in adult cerebellum, showed that Wiz was detected at the promoters of those with decreased expression but not at those with increased expression. This suggests that those sites at which reducing the levels of Wiz causes increased expression, i.e. Wiz is acting as a repressor, are indirect effects.

### Wiz might bind directly to DNA

While Wiz has no recognisable catalytic domains, it does have C2H2-type zinc fingers and these have an unusual widely-spaced configuration (*Matsumoto et al., 1998*). Most proteins that contain multiple zinc fingers have them in a cluster, with short 'linker' sequences (usually less than 10 amino acids) separating the fingers (*Schuh et al., 1986*). Most Wiz zinc fingers are at least 50 amino acids apart. The significance of the large linker configuration remains unclear (*Cléard and Spierer, 2001*). Other zinc finger-containing proteins with this configuration include Zfp292, Rlf and the *Drosophila* protein Su(Var)3–7 and all of these have been shown to bind DNA in vitro using electromobility shift assays (*Cléard and Spierer, 2001*; *Harten et al., 2015*; *Lipkin et al., 1993*). *Rlf*, like *Wiz*, was identified as an *E(Var)* in the *MommeD* screen (*Daxinger et al., 2013*). Little has been reported about the function of Zfp292.

Consistent with previous studies using different Wiz antibodies (*Matsumoto et al., 1998*; *Ueda et al., 2006*; *Ma et al., 2015*; *Simon et al., 2015*) we identified multiple isoforms of Wiz present in the mouse, three at ~100 kDa and one larger ~160 kDa. The requirement of these different isoforms is unknown and requires careful, isoform-specific study of Wiz. It is also possible that the three ~100 kDa isoforms represent covalent modifications to the Wiz protein. For example, it is known that G9A can methylate non-histone targets, including Wiz (*Rathert et al., 2008*).

## Wiz binding overlaps with CTCF binding in the cerebellum

Our ChIP-seq studies suggest that in adult cerebellum, Wiz binds at promoters and at sites which CTCF, a recognized transcriptional activator, is also known to bind. Because this study has been carried out in a tissue made up of more than one cell type, peak overlap does not necessarily imply that the two proteins bind together. It could be explained by CTCF binding to these locations in one cell type and Wiz binding in another cell type. Competitive binding with CTCF and the closely related BORIS has been demonstrated (*Pugacheva et al., 2015*). To our knowledge, Wiz has not been pulled down with antibodies to CTCF (*Xiao et al., 2011*; *Yusufzai et al., 2004*), though a protein-protein interaction is possible in neural cell types where *Wiz* is expressed highly and co-immunoprecipitation of binding partners remains to be tested.

CTCF has multiple roles in organizing the genome; it is enriched at topologically associated domain boundaries and can mediate interactions between an enhancer and promoter elements (*Bell et al., 1999*; *Ong and Corces, 2014*). Broad regions marked by H3K9me2 often contain small 'euchromatic islands' that are devoid of H3K9me2, are DNase1 hypersensitive and have CTCF binding (*Wen et al., 2012*). A recent study in a number of human cell lines found that reducing WIZ levels resulted in less G9a at CTCF-binding sites across the genome (by carrying out ChIP-seq for G9a) (*Simon et al., 2015*). Further work is needed to show mechanistically how CTCF, Wiz and G9a establish chromatin at these euchromatic islands.

ChIP-seq for WIZ in a human kidney cell line has recently been published (*Bian et al., 2015*) and no overlap with CTCF-binding sites was reported. An in-house antibody (not commercially available) was used. Approximately 11,000 WIZ-enriched regions were detected and the consensus binding site that emerged, CATTCCATTCCATT, is quite different from the one reported here. This sequence was found in only 4.5% of their peaks, whereas the Wiz consensus reported here (Wiz motif 1) was found in 70% of the peaks identified in our study. Remapping of their Wiz ChIP-seq dataset was carried out, and failed to produce the peaks reported except at a repetitive HSATII satellite element, suggesting a mapping artefact. The differences might also be linked to the different cell types used. Resolution of this discrepancy will require the publication of Wiz ChIP-seq datasets with additional anti-Wiz antibodies and in different tissues.

## Wiz regulates protocadherin gene expression

Protocadherin genes are expressed mainly in neurons (*Kohmura et al., 1998*; *Wu and Maniatis, 1999*) and here we report the involvement of Wiz at protocadherin β genes. CTCF has been shown by others to bind to the promoters and enhancer of this locus, mediating the looping required for the transcription of single protocadherin β genes in single neural cells (*Guo et al., 2015*). Given that almost all sites in the cerebellum that were bound by CTCF were also bound by Wiz, it is possible that Wiz has a function in chromatin looping between enhancers and promoters. Determining the importance of Wiz in looping will require chromosome conformation capture (3C) analysis in Wiz null cells. Early embryonic lethality precluded testing this in the tissue used in this study.

The protocadherin cluster exists as a large tandem array. At this locus, heterochromatinisation facilitates single isoform expression in single cells i.e. the silencing of the majority of protocadherin genes in any single cell is associated with heterochromatin formation (*Kawaguchi et al., 2008*; *Toyoda et al., 2014*); the expressed loci must remain active despite being embedded in heterochromatin and it is possible that Wiz has a role in this scenario. In keeping with this hypothesis, the RNA-seq data presented here suggest that other loci sensitive to Wiz haploinsufficiency include genes located adjacent to telomeres, well known heterochromatic regions.

## Wiz is required for normal behaviour in the adult mouse

The importance of epigenetic modifications in neural function is emerging; many enzymes that modify histones and DNA have been found to be critical for development and function of the brain. Prior to this study, no link between *Wiz* and behaviour had been shown in either mice or humans. Here we report that *Wiz*^MommeD30/+ mice showed a more anxious phenotype than wildtype littermates in a number of assays of anxiety. It is possible that heterozygosity for rare *WIZ* mutations in humans could influence neurological disorders that have a complex genetic aetiology.

It remains to be seen if the gene expression changes and behavioural phenotype in *Wiz*^MommeD30/+ mice correlates with altered neuroanatomy. Mice with reduced expression of

protocadherin β genes, due to knockout of CTCF (*Hirayama et al., 2012*) or the downstream enhancer (*Yokota et al., 2011*), have disorganized formation of the barrel cortex, a region of the brain important for processing information for the tactile somatosensory pathway (*Petersen, 2007*). It remains to be determined if this phenotype in the barrel cortex is recapitulated in $Wiz^{MommeD30/+}$ mice.

## Materials and methods

### Mouse strains

$Wiz^{MommeD30}$ mice were produced in an ENU mutagenesis screen, as previously described (*Daxinger et al., 2013*). The ENU screen was carried out using a *Line3* strain that is on an FVB/NJ inbred background, as described previously (*Blewitt et al., 2005*). *Line3* are homozygous for a multi-copy GFP transgene with human α-globin promoter and HS-40 elements and the $Wiz^{MommeD30}$ mutation was maintained by crossing heterozygous mice to *Line3*. The $A^{vy}$ allele arose from a C3H/HeJ colony, was backcrossed to C57BL/6J for at least 20 generations and was maintained on that background in the heterozygous state ($A^{vy}/a$). Genotyping was carried out as previously described for the $Wiz^{MommeD30}$ (*Daxinger et al., 2013*) and $A^{vy}$ alleles (*Rakyan et al., 2003*), primers are listed in *Supplementary file 5*. All animal work was conducted in accordance with the Australian code for the care and use of animals for scientific purposes, this study was approved by the Animal Ethics Committee of La Trobe University, project numbers 12–74, 12–75, 15–01.

### ChIP-seq and ChIP-qPCR for Wiz in cerebellum

Cerebellum tissue from 8-weekold wildtype mice were snap-frozen and sent to Active Motif (Carlsbad, USA) for ChIP, library preparation, sequencing and initial data QC. Two cerebellums were pooled per biological replicate. The rabbit polyclonal anti-Wiz (NBP180586, Novus Biologicals, Littleton, USA) was used for ChIP on two biological replicates and an input chromatin sample was made by pooling equal amounts from the two biological replicates. Sequencing was carried out for 75mer read lengths on the NextSeq 500 platform (Illumina, San Diego, USA). At least 30 million reads were generated per library. Read alignment to the mouse mm10 genome build was carried out using the Bowtie2 program with default settings (version 2.2.2) (*Langmead and Salzberg, 2012*) and peak calling was carried out using the MACS2 program with the settings '–down-sample –call-summits' (version 2.1.0) (*Zhang et al., 2008*). Peaks were considered significant if they had a p-value of $\leq 1 \times 10^{-20}$.

Heat plots and read sequencing density figures were generated using the seqMiner program (version 1.3.3) (*Ye et al., 2011*) using default settings. Heat plots were generated by subsampling all datasets to approximately 16 million reads and clustering was carried out using the seqMiner program using default settings. When calculating deep sequencing read density over gene bodies, genes of different lengths were scaled to an equal number and the read density was calculated for each of these by the seqMiner program. Genes were classified as active (n = 13,279 genes) or silent (n = 25,152 genes) based on an averaged raw read count of less than 50 from RNA-seq carried out in adult cerebellum tissue (see below), active genes accounted for approximately ~99% of read alignments in the dataset.

To analyse the overlap of Wiz ChIP-seq peaks and ENCODE dataset (*Shen et al., 2012*) peaks, the latter were first downloaded from the UCSC genome browser (*Karolchik et al., 2003*), datasets were 3′ trimmed so that all datasets contained reads of equal length and mapping and peak calling was carried out as described above. ChIP-seq datasets for CTCF and the histone mark H3K4me3 were analysed and in each case this consisted of two biological replicates and an input dataset. Overlap between two peaks was defined as the coordinates of two peaks directly overlapping at one or more bases in the genome. Motif discovery was performed with significant Wiz peaks (region summit ± 500 bp) using the MEME-ChIP (version 4.10.0) (*Machanick and Bailey, 2011*) and MEME suite programs MEME, DREAME, CentriMo and Tomtom with default settings. The publically available Wiz ChIP-seq datasets were downloaded from the GEO website (www.ncbi.nlm.nih.gov/geo/) and mapped to the human genome (hg19 build) as described above.

ChIP-qPCR was performed with the Anti-Wiz antibody (NBP180586) on three biological replicates by Active Motif and loci were amplified using primers designed by Active Motif at regions of ChIP-

seq enrichment indicated in the Figures. Enrichment was calculated by running qPCR reactions alongside known amounts of DNA to generate a standard curve. These are represented as binding events detected per 1000 cells.

## Data availability

The data sets supporting the results of this article are available in the NCBI Gene Expression Omnibus under the accession code GSE76909.

## Timed matings

Embryos were produced by heterozygous intercrosses ($Wiz^{MommeD30/+} \times Wiz^{MommeD30/+}$) and detection of a post coital plug was defined as E0.5 dpc. Dams were euthanized by cervical dislocation and embryos dissected into 1X PBS, tissues were either snap frozen with dry ice for later RNA extraction or lysed overnight for DNA extraction as previously described (*Morgan et al., 1999*). The sex of embryos was determined by PCR using primer sets annealing either at X or Y chromosomal loci, listed in *Supplementary file 5*.

## RNA-seq

RNA was purified from 8-weekold male cerebellum or embryonic brain (E13.5) using Triazol reagent (Life Technologies, Carlsbad, USA), as per manufacture instruction. RNA sequencing was carried out by the Australia Genome Research Facility (AGRF, Parkville, AUS) and at least 20 million 100 bp single end reads were generated on an Illumina HiSeq platform for each sample, from libraries generated using the Illumina TruSeq RNA Sample Preparation kit (Illumina, San Diego, CA, USA). An initial QC analysis was performed by AGRF. Sequencing reads were mapped to the mouse mm10 genome build using the program Tophat (version 2.0.11) (*Trapnell et al., 2009*), read counts for gene exons were extracted using the program htseq-count (version 0.6.1) (*Anders et al., 2015*) and differential gene expression was assessed using the R-package DEseq (*Anders and Huber, 2010*), as described previously (*Isbel et al., 2015*). Genes were considered significantly differentially regulated if they had a fold-change ($Wiz^{MommeD30/+}/Wiz^{+/+}$) of less than 0.7 or greater than 1.3 (shown in figures on a log2 scale) and an adjusted p-value of $\leq 0.05$.

## Western blotting for Wiz in embryonic and adult tissues

Protein lysates from embryos were prepared by homogenising tissue in ten volumes of urea lysis buffer, as described previously (*Daxinger et al., 2013*). These were quantified using a BCA assay (Thermo Scientific, VIC, Australia), separated on polyacrylamide gels (Bio-Rad, Gladesville, NSW, Australia) and immunoblotted with antibodies directed against Wiz (NBP180586, Novus Biologicals, Littleton, USA) and against γ-Tubulin (T5192, Sigma Aldrich, Castle Hill, AUS). Clarity Western ECL substrate was used for visualisation (Bio-Rad, Gladesville, NSW, Australia).

## Affinity purification and mass spectroscopy

Nuclear extracts were prepared from either pooled embryonic brains (E13.5) or adult cerebellum using the Nuclear Complex Co-IP Kit (54001, Active Motif, Carlsbad, CA, USA) as described previously (*Harten et al., 2015*; *Isbel et al., 2015*). For each sample, 500 μg of nuclear lysate was incubated with 2 μg of anti-Wiz antibody (NBP180586, Novus Biologicals, Littleton, USA) or anti-IgG antibody (sc-2345, Santa Cruz Biotechnology, Dallas, TX, USA) overnight at 4 degrees Celsius. Immunoprecipitations were carried out using bead-conjugated Protein G (Dynabeads 10003D, Invitrogen, Mount Waverley, VIC, Australia) as per the Nuclear Complex Co-IP Kit manufacturer's instructions. Eluted proteins were analysed via mass spectrometry at the La Trobe University Mass Spectrometry Facility, and the data were analysed as described previously (*Harten et al., 2015*).

## Telemetry monitoring of mice

Age matched male mice, 8–12 weeks old, were individually housed in standard polypropylene cages (14 × 30 × 13 cm) with shredded paper as bedding enrichment, ambient temperature was 30 ± 1°C, with a 12 hr light/dark cycle and lights-on at 07:00 hr. Mice received standard rodent chow and water *ad libitum*. After one week of acclimatisation mice were put under general anaesthesia and surgically implanted with a biotelemetry device (Mini-mitter, Bend, USA) into their peritoneal

cavities. After one week of recovery, the biotelemetry devices were used to monitor body temperature and locomotor activity. A platform receiver sampled the device's temperature (± 0.1°C) and location at one minute intervals continuously and this was decoded by a software package VitalView (Starr Life Sciences Corp, Oakmont, USA). Following a 12 day period of uninterrupted monitoring, mice were exposed to an elevated plus-maze (EPM) test of anxiety-like behaviour (see below). Immediately following the EPM test, the mice were returned to their home cage for approximately 120 min, then anesthetized using a lethal dose of pentobarbital sodium (Lethabarb, Virbac Pty. Ltd., Milperra, Australia). The telemetry data (activity and body temperature) were collapsed across the 12 days of testing to derive the activity for each hour in the 24 hr cycle. These data were then analysed using a repeated measures analysis of variance (ANOVA) with time as the within subjects factor and genotype and cohort as between subjects factors. A Greenhouse–Geisser correction epsilon ($\varepsilon$) was used to correct for potential violation of the sphericity assumption for the within-subject measure. Where significant main or interaction effects were found, post hoc Fisher's LSD tests were used to reveal pairwise differences between groups. As there was no main effect for cohort ($F_{(1,17)} = 1.313$, $p=0.268$) the data from the two cohorts were combined.

## Elevated plus-maze testing

Following the 12day monitoring period, mice were exposed to the EPM test. Each mouse was placed on the EPM for a duration of 5 min and behaviour was recorded using a digital video camera fixed above and overlooking the apparatus. Position and movement data was analysed using the Ethovision XT behaviour tracking software. The number of entries in the open arms, a common measure of rodent anxiety-like behaviour was analysed using an independent samples t-test.

## Light-dark behaviour testing

Light-dark behaviour testing was carried out on mice aged 2–6 months (n = 43) using methodology that was previously described (*Balemans et al., 2010*). Each animal was housed separately for at least 3 weeks prior to testing. Testing was carried out for a duration of 10 min and was analysed using an independent samples t-test.

---

## Additional information

### Funding

| Funder | Grant reference number | Author |
| --- | --- | --- |
| National Health and Medical Research Council | Australia fellowship, 1058345 | Emma Whitelaw |

The funders had no role in study design, data collection and interpretation, or the decision to submit the work for publication.

### Author contributions

LI, MWH, Conception and design, Acquisition of data, Analysis and interpretation of data, Drafting or revising the article; LP, Acquisition of data, Analysis and interpretation of data, Drafting or revising the article; HW, Analysis and interpretation of data; LD, EW, Conception and design, Analysis and interpretation of data, Drafting or revising the article; HO, Analysis and interpretation of data, Drafting or revising the article; AS, Acquisition of data; AJL, Drafting or revising the article

### Author ORCIDs

Luke Isbel, http://orcid.org/0000-0002-5270-4347
Alex Spurling, http://orcid.org/0000-0002-4368-6191
Emma Whitelaw, http://orcid.org/0000-0002-2320-2903

### Ethics

Animal experimentation: All animal work was conducted in accordance with the Australian code for the care and use of animals for scientific purposes, this study was approved by the Animal Ethics Committee of La Trobe University, project numbers 12-74, 12-75, 15-01.

# Additional files

## Supplementary files

• Supplementary file 1. Wiz genome wide binding sites. Wiz ChIP-seq binding peaks (mm10 genome build) identified in adult wildtype cerebellum (n = 2 Wiz ChIP-seq biological replicates, 1 input sample).

• Supplementary file 2. Genes significantly differentially expressed in $Wiz^{MommeD30/+}$ E13.5 male brain. The average read count from both $Wiz^{+/+}$and $Wiz^{MommeD30/+}$ animals (n = 3 biological replicates per genotype) are shown, with the fold-change ($Wiz^{MommeD30/+}/Wiz^{+/+}$) and adjusted significance value for each gene predicted to be differentially expressed. Read counts are normalized for library size.

• Supplementary file 3. Genes significantly differentially expressed in $Wiz^{MommeD30/+}$ E13.5 female brain. The average read count from both $Wiz^{+/+}$ and $Wiz^{MommeD30/+}$ animals (n = 2 biological replicates per genotype) are shown, with the fold-change ($Wiz^{MommeD30/+}/Wiz^{+/+}$) and adjusted significance value for each gene predicted to be differentially expressed. Read counts are normalized for library size.

• Supplementary file 4. Genes significantly differentially expressed in $Wiz^{MommeD30/+}$ cerebellum. The average read count from both $Wiz^{+/+}$ and $Wiz^{MommeD30/+}$ animals (n = 3 biological replicates per genotype) are shown, with the fold-change ($Wiz^{MommeD30/+} / Wiz^{+/+}$) and adjusted significance value for each gene predicted to be differentially expressed. Cells highlighted orange are protocadherin β cluster genes and cells highlighted green are cadherin 11-like cluster genes. Read counts are normalized for library size.

• Supplementary file 5. Oligonucleotides used in the study.

## Major datasets

The following dataset was generated:

| Author(s) | Year | Dataset title | Dataset URL | Database, license, and accessibility information |
|---|---|---|---|---|
| Isbel L, Prokopuk L, Haoyu Wu, Lucia Daxinger, Oey H, Spurling A, Lawther A, Hale M, Whitelaw E | 2016 | Widely Interspaced Zinc Finger Motifs, Wiz, binds active promoters and CTCF-binding sites and is required for normal neural function in the mouse | http://www.ncbi.nlm.nih.gov/geo/query/acc.cgi?acc=GSE76909 | Publicly available at the NCBI Gene Expression Omnibus (accession no: GSE76909) |

The following previously published dataset was used:

| Author(s) | Year | Dataset title | Dataset URL | Database, license, and accessibility information |
|---|---|---|---|---|
| Bian C, Chen Q, Yu X | 2015 | G9a, ZNF644 and WIZ ChIP-seq results | http://www.ncbi.nlm.nih.gov/geo/query/acc.cgi?acc=GSE62616 | Publicly available at the NCBI Gene Expression Omnibus (accession no: GSE62616) |

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
