## [Decision Letter]

Thank you for submitting your article "Wiz binds promoters containing CTCF-binding sites and is required for normal behaviour in the mouse" for consideration by *eLife*. Your article has been reviewed by three peer reviewers, and the evaluation has been overseen by a Reviewing Editor and Jessica Tyler as the Senior Editor. One of the three reviewers has agreed to reveal his identity: Michael Stallcup (reviewer #3).

The reviewers have discussed the reviews with one another and the Reviewing Editor has abridged their comments and drafted this decision to help you prepare a revised submission.

1) The antibody specificity has been questioned.

i) The Western blot (Figure 2—figure supplement 1) and the new Cerebellum Western showing the high mw band are convincing. Therefore, the new Cerebellum Western should be included in Figure 2—figure supplement 1.

ii) A point not being addressed is the ChIP quality/specificity of the antibody. Robust controls are required to demonstrate that the ChIP signals in this manuscript are Wiz dependent: Although Wiz null mice are lethal, 14d embryos have been published (Daxinger et al., 2013) and these embryos have been used for western in this manuscript. Therefore, the WizMD30/MD30 genotype can be and has to be tested by ChIP and qPCR at validating sites (i.e. about 10 positive and negative sites) and compared to wildtype. As a positive control, both genotypes have to be tested for H3 ChIP or similar at these sites. Data may be presented as [Supplementary-material SD1-data]. Alternatively, in case of technical limitations to "ChIPping" homozygous embryos, Wiz ChIPseq with chromatin of a cell line +/- siRNA against Wiz has to be done to document that ChIP signals are indeed dependent on Wiz. Other suggestions to test this include trying another Wiz antibody by Chip-qPCR or performing some reporter assays (e.g. luciferase assays to test promoter activity with and without Wiz overexpression). Regardless, some test for antibody validation should be performed.

2) Isbel et al. convincingly argue that the relevant chromosomal region shows a single mutation in the Wiz gene as identified by exome sequencing. Nevertheless, this neglects mutations outside of exons, which potentially may influence the expression of the 48 genes in this chromosomal region (Daxinger et al., 2013) and/or of non-coding RNA. The authors should discuss this potential combined effect of Wiz and of unknown genes.

3) The manuscript may be strengthened if less emphasis is put on the possible transcriptional activator function of Wiz. The transcriptional effects measured in E13.5 are a convolution of direct and indirect effects. Furthermore, only a handful of genes harboring Wiz binding in their promoter are affected, raising unanswered questions about the underlying mechanisms. With the present data it is difficult to conclude about a direct role of Wiz in transcriptional activation, especially in the absence of any biochemical/mechanistic exploration of the phenomenon. This should be commented on.

4) It is relevant to consider whether the occupancy of either protein at their shared binding regions depends on the occupancy of the other protein. How many of the differentially expressed genes are bound by Wiz and CTCF in their promoters, respectively? For the down-regulated genes, how many actually have Wiz/CTCF binding at their TSS? How many do not? Figure 6 are rather misleading. It may be more informative to plot the density per gene in a heat map as the one in Figure 3 (sorted either by RNA-seq fold change or CTCF/Wiz ChIP-seq signal), or at least include a Venn diagram showing the overlap of the number of down regulated genes and Wiz and CTCF binding at the corresponding TSS.

5) The title and Abstract are a bit misleading. In their current state they seem to suggest that Wiz specifically binds promoters containing CTCF sites, which is not the case as Wiz binds most CTCF sites as well as most promoters. I suggest rephrasing to reflect the data more appropriately (e.g. "Wiz binds active promoters and CTCF binding sites…". Same throughout the text).

---

## [Author Response]

*1) The antibody specificity has been questioned.*

*i) The Western blot (Figure 2—figure supplement 1) and the new Cerebellum Western showing the high mw band are convincing. Therefore, the new Cerebellum Western should be included in Figure 2—figure supplement 1.*

A large, ~160kDa isoform of Wiz has previously been reported in adult cerebellum (Matsumoto et al., 1998, Bian et al., 2015) and we have detected this with our antibody on a Western using protein extracted from this tissue but this was not included in the original version of the manuscript. This data is now included in Figure 2—figure supplement 1 and the following text has been added to the manuscript:

“A larger ~160 KDa isoform has previously been reported in adult cerebellum (Matsumoto et al., 1998) and consistent with that report a band of this size is detected with our antibody in protein extracted from this tissue (Figure 2—figure supplement 1).”

ii) A point not being addressed is the ChIP quality/specificity of the antibody. Robust controls are required to demonstrate that the ChIP signals in this manuscript are Wiz dependent: Although Wiz null mice are lethal, 14d embryos have been published (Daxinger et al., 2013) and these embryos have been used for western in this manuscript. Therefore, the WizMD30/MD30 genotype can be and has to be tested by ChIP and qPCR at validating sites (i.e. about 10 positive and negative sites) and compared to wildtype. As a positive control, both genotypes have to be tested for H3 ChIP or similar at these sites. Data may be presented as [Supplementary-material SD1-data]. Alternatively, in case of technical limitations to "ChIPping" homozygous embryos, Wiz ChIPseq with chromatin of a cell line +/- siRNA against Wiz has to be done to document that ChIP signals are indeed dependent on Wiz. Other suggestions to test this include trying another Wiz antibody by Chip-qPCR or performing some reporter assays (e.g. luciferase assays to test promoter activity with and without Wiz overexpression). Regardless, some test for antibody validation should be performed.

We are not able to ChIP Wiz in the developing embryo due to embryonic lethality. The 14 day old embryos mentioned above are grossly abnormal, small (Daxinger et al., 2013) and not suitable for ChIP. Unfortunately, primary cells cultured from these homozygous embryos have limited viability (not shown). To address the issue, we have used an unbiased tandem affinity purification + mass spectroscopy approach to purify the Wiz complexes in wildtype embryonic and adult neural tissues. We identified 45 proteins that were present in anti-Wiz antibody pull down samples and absent in anti-IgG antibody controls. Ranking these by the semi-quantitative number of peptides recovered from embryonic brain, where expression of Wiz is high, shows that four members of Wiz-Zfp644-EHMT1-EHMT2 complex are in the top five proteins identified in the dataset.

This mass spectroscopy data has now been included as [Supplementary-material SD1-data] and the protocol has been added to the Materials and methods section. The following text has been added to the manuscript:

“In addition, we used an unbiased tandem affinity purification approach and compared mass spectroscopy data with previously published datasets analysing Wiz binding partners. […] Four members of the Wiz-Zfp644-EHMT1-EHMT2 complex (Ueda et al., 2006, Bian et al., 2015) were in the top five proteins in the dataset, ranked by peptide count in embryonic brain (Figure 2).”

2) Isbel et al. convincingly argue that the relevant chromosomal region shows a single mutation in the Wiz gene as identified by exome sequencing. Nevertheless, this neglects mutations outside of exons, which potentially may influence the expression of the 48 genes in this chromosomal region (Daxinger et al., 2013) and/or of non-coding RNA. The authors should discuss this potential combined effect of Wiz and of unknown genes.

It is unlikely that mutations inside the linked interval but outside exons are responsible for the effects for two reasons:

A) The frequency of ENU mutations (~1 mutation/Mb, Daxinger et al., 2013) make this unlikely;

B) The absence of changes in the transcription of genes in the linked interval (excluding the Wiz gene itself). What was not mentioned in the previously version of the manuscript was the fact that no genes within the linked interval were miss-expressed in any of our mutant RNA-seq datasets. The following paragraph has been added to the Discussion:

“We previously described a *Wiz* mutation in the *MommeD30* strain (Daxinger et al., 2013). This was the only exonic mutation within the linked interval and in agreement with this being the causative *MommeD30* mutation, RNA-seq data presented here show no other genes within the linked interval were mis-expressed in mutants.”

3) The manuscript may be strengthened if less emphasis is put on the possible transcriptional activator function of Wiz. The transcriptional effects measured in E13.5 are a convolution of direct and indirect effects. Furthermore, only a handful of genes harboring Wiz binding in their promoter are affected, raising unanswered questions about the underlying mechanisms. With the present data it is difficult to conclude about a direct role of Wiz in transcriptional activation, especially in the absence of any biochemical/mechanistic exploration of the phenomenon. This should be commented on.

The following sentence has been modified in the Abstract:

“Wiz is generally considered a transcriptional repressor. Here we provide evidence that it functions as a transcriptional activator.” has been changed to “Wiz is generally considered a transcriptional repressor. Here we provide evidence that it may also function as a transcriptional activator.”

The following sentence has been modified in the Introduction section:

“*MommeD30* was identified in the screen because mice heterozygous for the mutant allele showed decreased expression of the GFP transgene, consistent with a role for Wiz as a transcriptional activator.” has been changed to “*MommeD30* was identified in the screen because mice heterozygous for the mutant allele showed decreased expression of the GFP transgene.”

The following sentence has been modified in the Results section:

“The effect of haploinsufficiency at the *A^vy^* locus is similar to the effect at the GFP transgene reporter, i.e. reduced levels of Wiz were associated with increased silencing, suggesting that the wildtype protein acts as an activator of transcription.” has been changed to “The effect of haploinsufficiency at the *A^vy^*locus is similar to the effect at the GFP transgene reporter, i.e. reduced levels of Wiz were associated with increased silencing.”

We have already commented that effects could be indirect and this remains in the Discussion:

“While any instance of decreased expression could, theoretically, be an indirect effect…”.

4) It is relevant to consider whether the occupancy of either protein at their shared binding regions depends on the occupancy of the other protein. How many of the differentially expressed genes are bound by Wiz and CTCF in their promoters, respectively? For the down-regulated genes, how many actually have Wiz/CTCF binding at their TSS? How many do not? Figure 6 are rather misleading. It may be more informative to plot the density per gene in a heat map as the one in Figure 3 (sorted either by RNA-seq fold change or CTCF/Wiz ChIP-seq signal), or at least include a Venn diagram showing the overlap of the number of down regulated genes and Wiz and CTCF binding at the corresponding TSS.

As suggested, we have changed the read tag density plots (Figure 6) to heat map plots that are generated from the same data and show ChIP-seq binding density for each gene. This shows a direct assessment of Wiz occupancy at each of the genes that are deregulated and is more transparent for the reader. The figure legend has been modified to reflect this.

*5) The title and Abstract are a bit misleading. In their current state they seem to suggest that Wiz specifically binds promoters containing CTCF sites, which is not the case as Wiz binds most CTCF sites as well as most promoters. I suggest rephrasing to reflect the data more appropriately (e.g. "Wiz binds active promoters and CTCF binding sites…". Same throughout the text).*

This point has now been addressed throughout the manuscript. The title has been changed from "Wiz binds active promoters containing CTCF binding sites…" to "Wiz binds active promoters and CTCF binding sites…". The sentence in the Abstract – “…Wiz peaks were found at promoters showing enrichment for the transcription factor CTCF” – has been changed to: “ChIP-seq was performed in adult cerebellum and Wiz peaks were found at promoters and transcription factor CTCF binding sites”. The sentence in the Discussion – “Our ChIP-seq studies suggest that in adult cerebellum, Wiz binds at promoters at which CTCF, a recognized transcriptional activator, is also known to bind” – has been changed to: “Our ChIP-seq studies suggest that in adult cerebellum, Wiz binds at promoters and at sites which CTCF, a recognized transcriptional activator, is also known to bind”.